psychology/behaviour/computer modelling and simulation

animal welfare, wellbeing, sentience, model, digital twin, precision livestock farming

**Author for correspondence:**
Sergey Budaev
e-mail: sbudaev@gmail.com

# Computational animal welfare: towards cognitive architecture models of animal sentience, emotion and wellbeing

Sergey Budaev[1], Tore S. Kristiansen[2], Jarl Giske[1] and Sigrunn Eliassen[1]

[1]Department of Biological Sciences, University of Bergen, PO Box 7803, 5020 Bergen, Norway
[2]Research Group Animal Welfare, Institute of Marine Research, PO Box 1870, 5817 Bergen, Norway

SB, 0000-0001-5079-9795; TSK, 0000-0001-5904-0224; JG, 0000-0001-5034-8177; SE, 0000-0001-6728-3699

To understand animal wellbeing, we need to consider subjective phenomena and sentience. This is challenging, since these properties are private and cannot be observed directly. Certain motivations, emotions and related internal states can be inferred in animals through experiments that involve choice, learning, generalization and decision-making. Yet, even though there is significant progress in elucidating the neurobiology of human consciousness, animal consciousness is still a mystery. We propose that computational animal welfare science emerges at the intersection of animal behaviour, welfare and computational cognition. By using ideas from cognitive science, we develop a functional and generic definition of subjective phenomena as any process or state of the organism that exists from the first-person perspective and cannot be isolated from the animal subject. We then outline a general cognitive architecture to model simple forms of subjective processes and sentience. This includes evolutionary adaptation which contains top-down attention modulation, predictive processing and subjective simulation by re-entrant (recursive) computations. Thereafter, we show how this approach uses major characteristics of the subjective experience: elementary self-awareness, global workspace and qualia with unity and continuity. This provides a formal framework for process-based modelling of animal needs, subjective states, sentience and wellbeing.

# 1. Introduction

Animal welfare has grown into an important interdisciplinary area involving significant public concern and societal influence [1–5]. Nonetheless, there are still controversies over the application of the well-being and welfare (the state of wellbeing [2]) concepts to animals [3,6–8]. Indeed, much-used definitions of wellbeing emphasize its multi-faceted nature and link it with subjective feelings, emotions and sentience (see glossary) [6,9–11]. Animal (including human) wellbeing is most naturally understood from the first-person perspective [10]. This makes analyses of subjective phenomena (see glossary) nearly unavoidable in this field [6,10,12,13].

Subjective feeling-based definitions of welfare are sometimes criticized for not being compatible with 'objective science'. Critiques claim that physical health, naturalness and similar elements deemed easily measurable should define what welfare really is [6,12]. Folk judgements of animal welfare seem to support this to some extent [14,15]. If animals cannot communicate their subjective experience, a sceptic view is to remain agnostic. This entails ignoring their existence, resulting in a utilitarian approach to animal welfare: the main reason why animals matter is what they provide to us as farm or companion animals [16]. Then, the study of subjective feeling can divert us in the wrong direction [16]. But deliberately stripping complex cognitive abilities and subjectivity out sets an unnecessary ceiling on our understanding of welfare in healthy animals and fails to address the concerns for the animal suffering that were at the inception of animal welfare science [6,11].

Further, strong agnosticism can be ungrounded [17–20]. There is growing evidence for fascinating cognitive capacities in animals, including intentionality [21], components of conscious experience [13,18,22–24], planning and thinking [25–27], episodic memory [28,29] and meta-cognition [30,31]. It can therefore be fruitful to analyse complex cognitive abilities and subjective phenomena if we want to understand animal behaviour [32] and welfare [6,13]. There are still significant controversies over whether subjective phenomena, self-awareness and consciousness are just epiphenomena [20,33], or have adaptive significance [34–37] and whether they can be used to account for human and animal behaviour. However, as they are the essence of animal feelings and suffering, they are critical for understanding wellbeing [6,11].

This view justifies the development of theoretical approaches to animal sentience and subjective processes beyond general philosophical and metaphysical thinking. A better understanding of the core concepts of wellbeing requires formal models for standardized assessment (e.g. [38,39]) and quantitative models [40] that reflect how physiological, cognitive and behavioural needs relate to subjective states, emotions, suffering and stress.

Our main aim is to develop new computational technology for animal welfare. We bring together ideas from animal behaviour, neurobiology and computational cognitive science. First, we outline a general organization of adaptive behaviour in animals focusing on subjective state and predictive cognition (§2). We then provide a brief review of the wellbeing concept and its links with subjective state and predictive cognition (§3). Next, in §4, we describe the essence of the computational modelling approach to cognition with a focus on cognitive architecture. We then provide a natural framework for process-based modelling of both subjective wellbeing, sentience and consciousness evolving through natural selection and bounded by the ecological conditions in §5. Finally, in §6, we propose a way forward towards the computational animal welfare science and outline several implications (§7). While our computational framework has been used in evolutionary ecology (e.g. [41,42]), this proposal for animal welfare is currently at an early stage. We think it is rich enough to inspire empirical and modelling work as well as a critique that will help to refine it in this area. We also believe that it may become an effective tool in the current era of digitalization and precision livestock farming [43,44].

# 2. How adaptive behaviour is produced

## 2.1. Animals as autonomous predictive decision-making agents

Animals live in complex and unpredictably changing natural landscapes, where making adaptive decisions require trading priorities among various needs related to survival, growth and reproduction [45–47]. This is a computationally complex task [48–50]. Rather than having automated responses to any immediate signal, animals have evolved to be 'autonomous agents' (see glossary) [51–53]. Their behaviour is generated endogenously [51], driven by the internal state [54] and follows from

expectations and goals set by the organism [52,53,55]. This has been long realized in ethology and animal welfare. The classical models of motivation [53,56,57] recognize the importance of internal causation. Similarly, the notion of behavioural needs in animal welfare was originally defined in terms of internal motivations [58,59].

There is a growing recognition of intrinsic spontaneity and indeterminacy of behaviour [60,61]. This spontaneity requires a capacity for predictive modelling in the nervous system that considers an animal's internal state [62–64]. Such predictive modelling is found even in insects [65,66]. Making predictions for the future is central to the adaptive functioning of the nervous [67,68] as well as hormonal system [69]. Thus, there is a growing realization that a stimulus–response paradigm is insufficient to account for complex behaviour [70,71]. Letting animals express their agency and cognitive complexity has therefore become an important animal welfare priority [55].

## 2.2. The link between decision-making and integrated self

Adaptive behaviour involves arbitration between alternative stimuli, responses and choices. A key mechanism for this is competition and negotiation among multiple neural ensembles and cognitive modules in brain function [72–74], learning [75], cognition [76,77] and decision-making [78–80]. The need to make goal-driven decisions with top-down behavioural control in dynamically changing environments requires the organism to maintain a view of itself and its surroundings through an internal (and hence, subjective) model [67,81,82]. This model may contain both evolved and learned components [50], should detect discrepancies from the reality [62,83,84] and monitor stress the state of wellbeing.

An animal can potentially produce a wide range of responses (including ignorance) to any given stimulus depending on its subjective model of its current state and its surroundings. Signs, like the smell of a predator or prey, are stimuli that signify something other than itself [85]. A multitude of signs can be found in the environment, some are also sent from the animal to other organisms voluntarily or involuntarily, and all organisms live in a 'semiosphere' of signs (and noise falsely interpreted as signs). Recognition of the meaning of important signs among the complex sensory input is essential for learning and prediction of the future. This recognition or interpretation capacity (semiotic freedom) is therefore also essential for how rich the animal's phenomenal world is and how well an animal copes with its environment. It will directly impact welfare in both positive and negative ways [85]. From a welfare perspective, we are most concerned with situations that pose a challenge to the animal. For instance, while wildebeest spot predators regularly and still keep on with their normal activity, there are combinations of signs that set the animal in alarm mode. When this occurs, cognitive, physiological and behavioural systems are focused in a single, unified defensive state often called 'fear'. Such global emotional and motivational states are of central importance for what is considered important (or irrelevant) for appropriate behavioural response [86–89]. Threat imminence is instrumental in fear: low-risk anxiety requires wide information integration governed by conscious feeling while panic at high risk substitutes consciousness with rapid innate automatic responses [90].

Emotion or affect represents a combined behavioural, physiological and cognitive state that primarily carries an individual organism's value of stimuli or context [91–93]. Emotion is closely linked with the assessment of rewards and punishments [89,94] and functions to focus the animal to avoid harm and obtain valuable resources [93–95]. Affective states come in degrees, which translates to the concept of arousal: an elementary neuronal process that activates many cognitive processes, emotions and behaviours. It affects both ascending and descending brain pathways and brings about alertness to a range of sensory stimuli, reactivity and motor activity [96,97]. This leads to the circumplex model of affect in two dimensions: positive versus negative valence and low versus high arousal [89,98,99]. While initially developed in the context of human psychology, this view is supported by evolutionary models [100] and has been applied to animals [89,101–103].

Historically, the global state of the organism was introduced to neurobiology at the beginning of the twentieth century by Ukhtomsky [104,105] as the principle of the dominant. The dominant was thought of as the prevailing source of excitability in the nervous system that greatly influences responses of the organism to a wide range of stimuli at a particular time. The dominant was conceptualized as the basic mechanism for focusing attention, subjective model and anticipation of the near future [105]. In modern terms, the global emotional and motivational state reflects the arbitrated primary need state of the organism that defines its current behavioural goals and predicted future [50,106]. If the organism anticipates that it cannot satisfy its needs, it may become stressed [89,107].

Given a particular dominant state, the animal selects the appropriate action. Organisms of many species (at least those with motility and complex behaviour) have evolved the ability to make

predictions of their near future based on their internal model and use this to decide its next behaviour [50,89]. In the process, the internal model is challenged and updated with new information that appeared as a consequence of the behaviour. Prediction error—the mismatch between the expected and perceived information—is central for cognition based on prediction. New information resulting from the behaviour can modify the internal model, bringing about a continuous flow of goal-driven computations for predicting the best behavioural action onwards. Alternatively, the animal can try to keep the outside world to agree with the subjective predictions as much as possible, as in the active inference paradigm [64,68,108]. In this perspective, animals can be viewed as prediction machines [62,64,109] that have the ability to consider and forecast future 'bodily feelings' (emotions, tastes) that result from potential actions [89,91]. Examples come from associative learning [84,110,111], goal-oriented cognition [62,83] and sensorimotor control of action and behaviour [67,83,112]. In this view, motivation, emotion and wellbeing can be central components and provide the common currencies for prediction-based cognition [89,113].

The mechanisms in the pathways from genes to development, physiology and behaviour [114–116], and from perception to decision and action [117–119], are to a large degree modular throughout the Tree of Life. This means that the whole system can be decomposed into discrete functional and/or structural components. This can apply to emotions and motivations. For example, there are certain core types of affect [86,93]. Threat imminence is also thought to evoke distinct modules of fear [90]. There are still debates in what respect and to what degree human cognitive architecture is modular (e.g. [120–123]). Modularity can increase the functional efficiency of large networks [124], especially when we consider connection costs [115]. Modularity can significantly increase the efficiency and the ability of both the living organism and the evolving gene pool to cope with environmental change [125,126]. It allows components to be modified, duplicated, replaced or deleted without catastrophic loss of function to the whole organism [127]. In this way, modularity creates a potential for individual variation [42,128], facilitates adaptive evolution and increases evolvability [41,115,127].

Many animal brains have a mechanism ensuring widespread information access across multiple processes working in parallel [129]. General components that link numerous modules are crucial for the cognitive function. For example, broad neuronal communication across encapsulated modules provides a computational advantage [130]. In humans, awareness counteracts modularity through the maintenance of widespread, almost global, connectivity [131]. For consciousness, this mechanism has been called the dynamic global workspace: a functional hub that binds and propagates neural signals across a wide range of specific networks [132,133]. This workspace is likely to have evolved gradually, so that many animals have 'not-so-global' dynamic workspaces [134]. We will still call them global in this article, in the sense that they represent all that is connected. In vertebrates, convergent neuronal pathways integrating several projections are often found outside of the cortex [135]. For example, the habenula links many diverse circuits [136], and there are links integrating telencephalon with the cerebellum [137]. Converging connections are common not only in vertebrates but also in animals with small nervous systems [138,139].

Subjective experience cannot be found in a neural system based on reactive, feed-forward circuit organization alone [140–142]. Neither is compositional and computational complexity in itself sufficient for the subjective experience [140]. Subjective experience and consciousness arise in systems that are able to (i) integrate information, (ii) monitor itself, and (iii) generate and process virtual (hypothetical, possible) rather than actual information [141–143]. The simulation theory of cognition [144] accounts for the emergence of the subjective world through the development of integrated sensory-motor circuits. The whole circuit is activated when producing the next behavioural or physiological action. But it is also engaged (subjectively simulated) when the same action is planned, anticipated or even observed [145–148]. In humans, reactivated sensorimotor circuits are involved in conceptual processing, declarative knowledge and understanding [145,149]. Thus, the first-person conscious subjective experience involves the acquisition and re-entrant activation of linked sensorimotor and affective circuits [145]. Re-entrant here means repeated, recursive, activation of the same neuronal ensembles and circuits [132,150].

Emotions and personality are closely associated with subjective phenomena and self [88,93,120]. Consistent personality variation, linked with affect, exists in many taxonomic groups and can originate from shared genetic, physiological, developmental, neurobiological and cognitive mechanisms as well as an evolutionary adaptation [151–153]. Personality in humans can significantly depend on cognitive architecture [154]. From this, we can see that a unitary understanding of the integrated cognitive and behavioural phenotype—the self—emerges. It includes subjective processes as well as externally observable traits, personality. In this perspective is subjective wellbeing a crucial component of the system, because it links with information integration and self-monitoring.

# 3. Wellbeing and suffering: objective and subjective

Wellbeing describes what is good for the individual [10]. The notion of 'good' is here intimately linked to Darwinian fitness. Evolutionary forces have formed sensitivity to stimuli, emotions and availability of response mechanisms for the organism through random mutations and selective survival and reproduction. Thus, wellbeing is about how life is going on for the organism from its own perspective. There is, however, a range of views on how animal wellbeing can be defined scientifically. Some schools tend to focus on the healthy, stress-free physiological functioning of the organism, its ability to cope with the current environment, and satisfy its basic motivational and behavioural needs [155–157]. Naturalness, i.e. ability to perform natural, ecologically adaptive behaviour, is also crucial [15,16,158]. For others, welfare largely involves subjective cognitive needs [55,159] and feelings (e.g. 'Let us not mince words: animal welfare involves the subjective feelings of animals' [160, p. 1]). This view depends on the assumption that many animals are capable of various degrees of sentience: the capacity to experience feelings such as pain and suffering [6,12,13,161]. The common position in the field has become to understand animal wellbeing as a complex set of phenomena involving cognitive constructs, subjective awareness and desires in contrast to simple physiological and health status [6,12,162].

While the role of positive emotional states and pleasure has been emphasized in animal welfare [163–165], wellbeing is elusive and multi-faceted and difficult to define precisely. It is especially hard in species that do not frequently display positive emotional states, social and play behaviour. However, deficit in wellbeing—physiological and psychological stress and suffering—may be easier to define, detect and measure [3,166,167]. Stress is understood as an organism's response to the actual or predicted threat, challenge or disruption of the organism's homeostasis. Usually it involves an increase of the general arousal, activation of the autonomic nervous system: hypothalamic–pituitary–adrenal axis [168–170] in mammals or hypothalamic–pituitary–interrenal in fish [171,172]. Incidentally, invertebrates have specific stress hormones [173], e.g. hyperglycaemic hormone in crustaceans [174]. Stress is also a subjective state of perceiving potentially adverse changes [170]. Theoretical discussions emphasize complex cognitive and behavioural aspects of the stress involving the animal's inability to anticipate the kind of challenge (unpredictability), its extent and possible avoiding strategies (uncontrollability) in addition to simple physiological responses [107,170,175]. Thus, stress is intrinsically linked with the emerging predictive cognition paradigm where the organism is depicted as a prediction machine [62–64]. Stress as a response is closely associated with a range of negative internal emotional states. In the animal wellbeing literature, such states have traditionally been subsumed under suffering. Suffering is defined as 'intensely and/or enduring unpleasant subjective feeling' [59, p. 210] or a 'wide range of unpleasant emotional states' [160 p. 1] or 'unpleasant feeling, which is prolonged or severe' [176, p. 374]. Thus, wellbeing is a combination of both objective and subjective aspects of the organism and reflects not just its current state, but also the anticipated change for better or for worse [9,159,177]. The needs of the animal (whether objective, such as adequate food and health, or subjective), its current state and anticipation of the near future are central for understanding wellbeing.

# 4. Computational models of cognition and cognitive architecture

Many biological and especially brain processes are computational, at least in the generic term [178,179]. Computation generally means manipulation of specific elementary units (e.g. digits, strings, neural spikes, continuous physical processes, variables, etc.) that are defined according to rules independently of the physical media that implement them [180]. Computation describes inherent functional and 'algorithmic' processes at the level of molecules and reaction networks [181,182], neural networks, the whole brain [183,184] and up to higher forms of cognition [185,186]. In this perspective, cognition and even the human mind (see glossary) are to a significant degree accounted for by computation [185,187,188]. Computational modelling and simulation of perception, cognition, learning, motivation, emotion and mind are therefore among the most natural ways of understanding the brain function and behaviour [70,183,185].

Complex computational models of the human mind are often implemented in the form of cognitive architecture [189–191]. This is a general framework that can be used through building more detailed computational models of cognition and behaviour focusing on particular problems [192,193]. It refers to algorithmic models of brain functioning rather than structural descriptions of neuronal wiring and brain morphology. It also differs from simple elegant mathematical equations, such as a function linking reinforcement rate and the response rate. Typically architectural models work through

building a virtual agent mechanistically implementing cognitive and behavioural functions that can just 'run' [194]. This is a thriving research field integrating psychology, neurobiology, artificial intelligence, computer science and philosophy of mind [189,190]. Cognitive architecture also provides a valuable tool for building quantitative models of animal cognition, sentience and welfare [50,106].

# 5. A modelling framework for subjective wellbeing and behaviour

Over the years, we have developed a model of an adaptive architecture for decision-making [41,42,45,128,195–198]. It currently contains a general framework and computer software components [199,200] for building simulation models that integrate cognition and behaviour in a phenotype that includes genome, physiology, hormonal system, perception, emotions, motivation and cognition. It also includes a virtual environment where the agents live and the gene pool evolves. This *adapted heuristics and architecture* (AHA) [50,106] provides a methodology for computational simulation of various internal subjective states and processes that account for animal wellbeing. Characteristics of the AHA cognitive architecture evolve through natural selection: the genetic algorithm leads to evolutionary adaptation of individuals and the gene pool [197,201,202]. This aligns well with the common view that all the mechanisms that account for animal wellbeing, including subjective feelings, are Darwinian fitness adaptations [176,177,203]. The AHA modelling framework can tackle elementary computational mechanisms thought to underlie subjective experience in different animals. We argue that our common evolutionary history suggests a continuity in neural, computational and evolutionary mechanisms that underlie subjective phenomena [91,134,204]. We are therefore concerned with functionally defined concepts in the same way that founders of ethology [205] and comparative psychology [206] used various human-derived terms without anthropomorphizing them.

## 5.1. A brief outline of the *adapted heuristics and architecture* cognitive architecture

Analyses of wellbeing take the viewpoint of the individual animal: its individual state, needs and responses [6,160]. Thus, the AHA cognitive architecture [50,106] starts with the needs of the organism. The animal has the basic energetic needs to sustain its life as well as such needs as to avoid predators and secure reproduction, all adapted by the evolution of the gene pool towards a defined (but variable) environment.

The organism is continuously exposed to various signals from the external and the internal environment. How all the numerous sensory stimuli are selected and integrated is defined by the subjective internal model (SIM), which is the animal's image of itself and its surroundings (figure 1). The strengths of the causal factors that integrate specific classes of sensory input represent the different motivations. For example, perception of stimuli linked to food and the individual's gut defines hunger, while the perception of stimuli from predators defines fear. As mentioned for the wildebeest earlier, there are many normal situations where no signals from the body or the environment indicate an upcoming urge to the SIM. The animal can then attend to several types of motivations more or less simultaneously, but with lower efficiency, and without evoking wellbeing issues.

Motivational modules will compete for priority if there are more than one of them activated at any time. The strongest becomes the dominant emotional state of the organism, its global organismic state (GOS) [207] (it is a common assumption that the animal performs one major activity at a time [53,86]). This global state reflects how the organism trades priority across its various needs and selects the currently most important one given the state of its SIM. The strength of this dominant need is indicated by the level of the general arousal. It is conceived as one of the main wellbeing indicators, defining the severity of stress and negative feeling. The whole pathway from perception through motivation to the global organismic state, that integrates multiple sources of information, is called the survival circuit [86]. The AHA organisms have several such circuits that encapsulate different emotional states: fear, hunger, thirst, reproductive drive etc. (figure 1). The GOS and the arousal jointly determine attention: which stimuli the organism is going to use or ignore and to what extent in its ongoing decision-making process.

This top-down (or goal-driven) attention is produced endogenously when specific information (e.g. specific types of stimuli) is actively sought out from the external or internal environment, e.g. based on the SIM memory template. The GOS actively limits the animal's attention to information that is most relevant to the current functional and behavioural state of the organism [76,112,208]. The GOS and the arousal determine how the animal makes decisions and selects behavioural actions. The organism can predict its likely effects on the environment and itself via subjective simulation of the

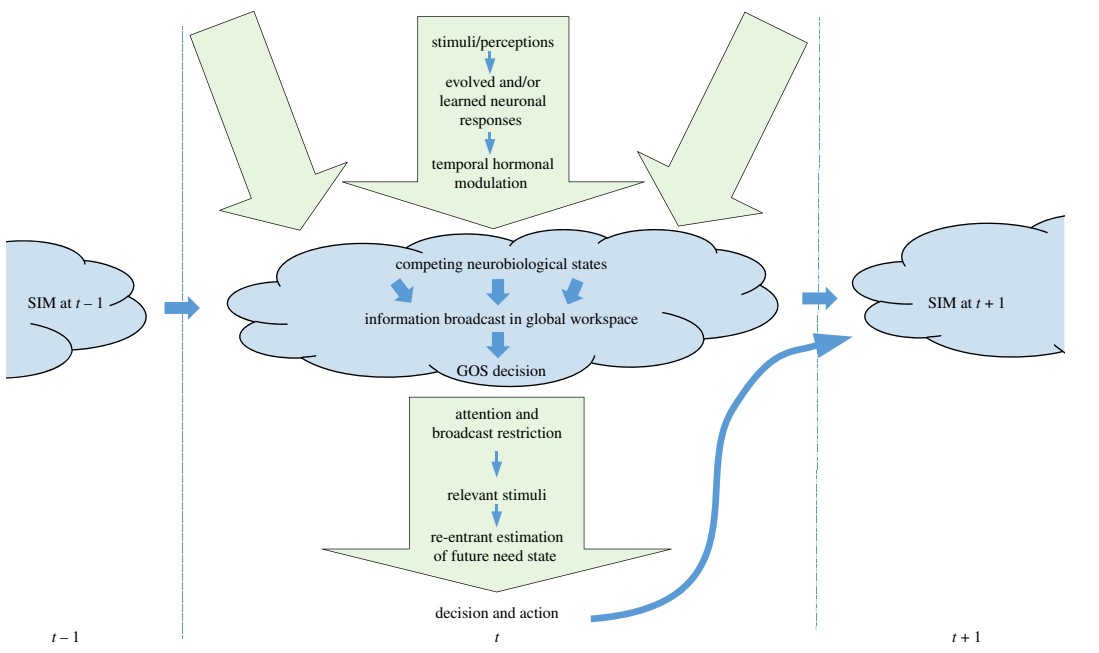

**Figure 1.** Major pathways for the subjective internal model (SIM) and global organismic state (GOS) over time ($t - 1 \rightarrow t \rightarrow t + 1$), where new sensations may enter the (blue) global broadcast area and then modify the behavioural action. The figure shows three competing (green) survival circuits in the appraisal phase and one winner in the action phase. Subjective feeling results from the dynamics of internal re-entrant activation of SIM for prediction-based decision-making and action-selection. Stress may become the long-term effect of a persisting challenge that the animal predicts will remain unresolved. The main factors not shown are the need state and prediction error. See text for explanation.

expected emotion and arousal as a common internal currency. This represents a simple mechanistic model of subjective feeling. The SIM and the GOS jointly represent the global workspace that defines the unitary subjective state of the individual (figure 1). In this perspective, the ability of the SIM to represent important aspects of the internal and external environment result from ongoing computation performed by (i) natural selection on the population gene pool (evolutionary adaptation) and (ii) by animals through individual learning. The framework is generic and allows the researcher to combine these elements in models of varying complexity. For example, one can implement a purely reactive stimulus–response organism, an organism adding a single re-entrant component (single kind of experience) linked with one GOS, or a much more complex system including re-entry for all GOS and global workspace. This would allow to depict different species and/or build simpler or more complex models for different purposes.

## 5.2. A model of subjective phenomena and elementary self-awareness

As outlined above, wellbeing is a substantially subjective (first-person) phenomenon that intimately depends on self-assessment and elements of consciousness. Direct experimental analysis of subjective feelings and experiences in animals is a daunting task. However, the use of models could help generate hypotheses and direct empirical research. The AHA cognitive architecture provides a mechanistic model for elementary forms of subjective phenomena that satisfies many definitions of sentience and consciousness.

At the simplest level, top-down attention generates a simple mechanism for both goal-directed behaviour and subjectivity. Since the GOS affects perception through the top-down selective attention, two organisms with identical genomes placed into identical environments, but differing in their current state (e.g. one slightly afraid, the other very hungry) will perceive their internal and external environments differently. Their whole pathways from perception to cognitive processing and behavioural action will be unique. Differences between the individuals accumulate over time, thereby causing diverging cognitive and behavioural trajectories. Internal parameters of these processes are inseparable from each organism (in fact, from the history of the organism–environment interactions). This satisfies our broad definition of subjective phenomena.

Elementary self-awareness is an important aspect of prediction-oriented cognition and action selection implemented in the AHA architecture. It is defined as 'the ability of the agent to assess its own internal state and use this information for decision making and action selection' [106]. It is implemented through re-entrant activation of the survival circuits: internal simulation of the organism's own potential actions (or the actual actions on an animal it observes) allows to determine the option that would result in the lowest arousal for negative emotions (or highest, for positive emotions). This links wellbeing with prediction error monitoring [62,64,84]. Consistently small prediction errors may indicate that the animal copes well enough with the environmental challenges. Furthermore, self-representation and self-monitoring are the most fundamental characteristics of consciousness [24,109,141].

This simple subjective simulation machinery reflects the emerging view that simpler forms of feeling, sentience and awareness not only have continuity across species [18,19,134], but can be found even in species with simple nervous systems [209]. The AHA cognitive architecture implements subjective experience by the mechanism of re-entrant activation, which is thought also to underlie human conscious experience [132,150,210].

The AHA architecture includes the two main components of conscious computations [141]: (i) self-monitoring in the form of elementary self-awareness and (ii) global workspace in the form of a unitary set of parameters for the SIM that hold the same for both producing and predicting inferred constructs (motivations, GOS). Several philosophical approaches to the nature of consciousness agree that its most important concept is *qualia* [20,211,212]. These are private, cognitively closed experiences that cannot be easily conveyed to others. Their main properties are *unity* and *continuity* [211,212]. Intriguingly, the AHA architecture provides a simple representation of qualia. Indeed, the way perceptual information is integrated to produce the internal state is unique for each individual and results from its previous history of interactions with the stochastic environment. It is *ineffable*: essentially non-inferable by an external observer without access to the organism's internal data structures. The great number of the external and internal stimuli that are uniquely filtered by attention in each case and potential processing pathways (especially if such pathways can be activated recurrently) makes it impossible to derive the internal data and cognitive structures even if the observer can record (the only observable) behaviour, input and output. Even for simple finite state machines, Moore's theorem [213] states that no amount of observation is sufficient to uniquely identify the machine. This provides a very simple model of *qualia*. The *unity* of individual experiences is based on the involvement of the same neuronal parameters for both generation and recurrent prediction of the arousal and the GOS [106]. The GOS also depicts a single central state affecting attention, cognition and behaviour at the next time steps. The *continuity* of the experience in our model is strengthened by a mechanism that tolerates small fluctuations of incoming stimuli to avoid fast GOS switching [106, p. 53].

The AHA architecture contains all the characteristic features of consciousness defined by Ginsburg & Jablonka [20,214]. (i) A *flexible value system* is represented by the GOS-linked top-down attention focus that modulates the goal depending on the internal state. (ii) *Unity and diversity through sensory binding* is implemented by the linking and integration of information between specific classes of stimuli and survival circuits. Additionally, GOS is a single unitary state. (iii) The *global availability of information* is again represented by the unitary GOS (G is for global) and by the use of the same set of parameters both for producing GOS and re-entrant simulations involved in the generation of the future action. The SIM is both global and persistent. (iv) *Temporal thickness* is equivalent to the above continuity of experience. It is implemented through an arousal-dependent motivation competition mechanism avoiding very fast switching between different GOS and behavioural states [106]. (v) *Selection* is implemented through internal motivational competition among several alternatives that gives rise to a particular GOS as well as top-down attention effects of the current GOS. (vi) The autonomous goal-driven behaviour of the AHA agent that depends on its GOS at each time point ensures *intentionality*: GOS also modulates further goal-driven actions through top-down restrictions of attention to new information. Finally, (vii) *self and embodiment* are ensured as AHA models each agent as a complete virtual organism including unique genome, physiology, cognitive architecture and behavioural actions. The agents 'live' in, and the evolving gene pool adapts them to, their virtual environment that contains food, predators and other agents as well as other factors the researcher may deem important for the animal and the model. Individual learning can be added on top of this, as it is of course important for adaptive behaviour in a range of animals.

We hypothesize that the subjective wellbeing can be a fundamental component of the evolved cognitive architecture because it provides the central vehicle for self-monitoring, with emotions

serving as an internal currency. Self-monitoring along with virtual processing and global availability are major elements of conscious processing. In effect, this will integrate the organism's needs, motivation, emotion and subjective feeling.

The model in figure 1 is both an abstraction and simplification of the processes in the brains and bodies of animals during decision-making. Each neuronal response function [42,195] and hormonal modulation [69,128] represents the net aggregate of a range of processes, and these aggregates are therefore not in themselves observable. Relevant parameter values can be found by evolving populations of digital organisms [196,215,216] in environments that resemble the evolutionary history of the species [42,217,218]. With this tool, we can theoretically investigate the behaviour and well-being of animals that live in a particular environment or those that are transferred to a novel situation or subjected to specific treatment procedures.

## 5.3. Model expectations

A model provides a theoretical framework that facilitates thinking about a phenomenon under study [219]. Simulation experiments can be performed that are not possible on living organisms. The cognitive architecture brings together basic building blocks from diverse fields to a complete machinery that can 'run'. This allows to study subjective processes that are highly relevant for understanding and ultimately improving animal wellbeing. Even the conceptual and graphical version of the model addresses important welfare challenges both for wild and domestic animals. Animal subjective states and wellbeing are expressed in observable behaviour: patterns and biases of decision-making and actions. Our cognitive architecture encompassing sensing, SIM, multiple competing survival circuits, global broadcast and the GOS determining top-down attention control (figure 1) can suggest certain behavioural patterns.

### 5.3.1. The animal's response to a stimulus depends on its global organismic state

While classical state-dependent theories in behavioural ecology [220–222] describe behaviour as dependent on states such as fat reserves or territory size, the key theoretical construct of the cognitive architecture is that the GOS is the decisive internal state of the organism. It determines which survival circuit a stimulus will be processed through and which behavioural response is finally evoked. Some stimuli can be linked with different survival circuits, for instance may conspecifics be judged as unwanted competitors for food or mates and wanted protection against predators. This is consistent with the neurobiological evidence that the animal's internal state can determine whether, for example, a zebrafish responds to a specific stimulus with approach or avoidance [223,224]. Even the same neurons could be involved in diverging responses under different perceived risk [225]. Controlling the animal's GOS and the arousal level are the main ways to achieve satisfactory subjective wellbeing.

### 5.3.2. Simultaneous pressures may lead to stress

This expectation concerns the situation when an animal simultaneously and over some time faces pressures from two or more survival circuits for control of the GOS. Frequent switching of the global state and attention without substantial reduction of the arousal would indicate inefficient decisions. High levels of neuronal activity are energetically costly [226,227] and could lead to neurotransmitter exhaustion [228]. High simultaneous recruitment of more than one survival circuit would normally translate to poor wellbeing. This agrees with the evidence that animals often try to avoid situations with many choices [229,230]. An ecological example is a series of studies by Milinski and Heller on sticklebacks [231,232]. When starved fish were exposed to food, they first prioritized feeding at a high rate in the centre of a prey swarm although they simultaneously received signals of imminent high predation risk. After short, they moved to the periphery of the swarm where feeding was both less efficient and less cognitively demanding. We interpret this as a conflict between two life-threatening factors, starvation and predation, controlled by separate survival circuits. As fish prioritized feeding, its hunger arousal fell while its fear arousal kept growing. The solution was to move to a place where it could attend to both survival circuits, but with lower efficiency. While this option was available by moving less than 1 m for the sticklebacks, it may be far less available for many animals. We interpret this as avoiding the recruitment of both hunger and defence circuits that would worsen subjective feeling and lead to stress. Parenthetically, it is instructive that similar considerations are discussed in the robotics literature [233].

### 5.3.3. Uncertainty would increase behavioural heterogeneity, but not at high arousal

In our framework, global information broadcast across several survival circuits is involved in the prediction of the best behaviour in the nearest future. The animal does this on the basis of 'what would it feel' if each of the available decision options is made. If there is high uncertainty as to the outcome of potential actions but the need state is not very strong (low to average arousal), one of many available survival circuits could be engaged, evoking diverse behavioural actions. This would increase the diversity and complexity of the behavioural output. However, in a situation of high need (high arousal), top-down attention control would significantly block all the stimuli not associated with the current GOS, leading to reduced behavioural complexity. Similar patterns have been documented, with stress reducing the diversity of behaviour in mammals [234–236].

### 5.3.4. High need state and stress may cause ambiguity bias

The situation of high need that is not satisfied for a long time maintains the arousal at a high level and upholds the relevant survival circuit's command of the GOS. This results in a narrow top-down attention span that will effectively ignore or suppress signals associated with all alternative survival circuits. Thus, a significant recognition and response bias towards the currently activated GOS may be expected. For example, an animal that remains hungry may interpret ambiguous stimuli and contexts as signals of food and respond accordingly. Similarly, an animal in chronic anxiety may display a negative cognitive bias by interpreting ambiguous stimuli as signals of danger or punishment. This tends to agree with the observations that many species display negative cognitive biases under stress [89,102,237–239].

### 5.3.5. Prolonged engagement of a single survival circuit and global organismic state may facilitate spontaneous change and (irrelevant) displacement activity

If the GOS is controlled by a single survival circuit for a long time (because the need is not satisfied), it may become subject to spontaneous dissipation of arousal due to neuronal exhaustion and neurotransmitter expenditure [106,240]). Then, a different survival circuit has a chance to win the competition over the next GOS. Because such dissipation would be accompanied by broadening of the attention span, different survival circuits could win and engage as the next GOS, even one that is irrelevant to the current needs. Thus, a displacement activity may appear. However, if the original motivational need is still not satisfied, it will be recruited again. Thus, short displacement activities may be expected to separate longer periods of behaviour that is unsuccessful at reducing the arousal of the GOS. Such a pattern—higher occurrence of displacement activities separating longer motivational states—has indeed been observed in primates [241] and honeybees [242].

## 6. Computational animal welfare: the digital twin approach

Aided by the recent progress in describing physiology, brain function and behaviour in mathematical and computational terms, simulation modelling has become a common tool in medicine and psychiatry. For example, 'computational psychiatry' combines big data analysis methods with theoretical models that account for mental illness as dysfunctional computations of the human cognitive system [243,244]. Animal welfare science could benefit from a similar computational approach. We argue that computer simulation of the animal's cognitive and behavioural function could provide a valuable tool to understand, monitor and improve the wellbeing status of animals in production facilities, those kept as pets as well as wild animals subject to various anthropogenic effects.

We expect that the developing computational animal welfare field should follow the emerging *digital twin* paradigm. Broadly, it involves a digital representation of a physical object through a computer-aided design and computer simulations [245]. The concept has recently been extended to biomedical engineering [246], agriculture [247] and even global climate [248]. It is considered a viable paradigm for personalized medicine [249]. The digital twin framework is a 'disruptive trend that will have increasingly broad and deep impact over the next five years and beyond' [250]. In our opinion, animal welfare could benefit from making use of this paradigm, especially because both health and well-being could be redefined in individual terms, i.e. in terms of the subjective state of an individual animal rather than the species or population norm.

# 7. Consequences for behaviour and welfare

While this computational framework is still in development, it has several important implications for how we think about animal behaviour and welfare. Computational models enable us to study specific hypotheses about the mechanics of animal cognition and behaviour before testing them experimentally. Further, challenging cognitive architecture models with new empirical data can generate novel hypotheses [219]. Even informal analyses of cognitive architecture can suggest interesting behavioural hypotheses.

The cognitive architecture models are theory-inspired, but there is a great potential in combining them with machine-learning approaches (e.g. [251]) based on big data collected in realistic (e.g. farm) settings. This is especially important at this era of digitalization in agriculture (and aquaculture) and precision livestock farming [43,44] that heavily depend on models and simulations [252]. Hybrid modelling approaches will result in improved transparency and accountability for decision-making [253,254] that is crucial for animal welfare [255]. Architectural models that implement the whole integrated phenotype [256] may be used to help monitor both the physical health and subjective well-being of animals. Such models may suggest good proxies for experimental assessment [257] and continuous monitoring [258] (e.g. using video [259]), predict indicators of deviations from good well-being state (based on indices of behavioural complexity [236]). And it can be used to run scenarios to forecast likely effects of environmental or procedural changes on animal health and the welfare status, including complex and emergent effects. In an advanced precision farm environment, a digital twin simulation can help predict various stress and welfare effects for both planned operational changes and possible perturbations. This would be cheaper and faster than using physical experimental systems and avoid unwanted animal welfare issues in the R&D process. The computational digital twin paradigm could prove useful also in the conservation ecology context. For example, it can predict effects of anthropogenic environmental interventions on the welfare of wild animals in semi-natural habitats, advancing the ecology of emotion (see [42,260–262]).

# 8. Concluding remarks

Consumers, legislators and representatives for the food production industries have a growing concern about animal welfare [2,3,7]. The welfare status of a production facility can be monitored via physiological and behavioural proxies to inner cognition, emotion and feeling that are expected to correlate with the welfare status [257,263]. We make a case for this method being complemented by computational animal welfare models.

Most researchers will agree that animal wellbeing is intrinsically linked with subjective phenomena in the broad sense, which may or may not include conscious states similar to those experienced by humans [6,9,11]. The recent decades have witnessed significant progress in neurobiological and neurocomputational mechanisms of human conscious experience [132,141,264,265]. However, we are still quite far from understanding the phenomenal consciousness and brain mechanisms of cognition in general [266]. In our opinion, one way to progress is to use functional definitions of subjective phenomena, just as the founders of ethology used to do for behaviour [53,205]. In the empirical animal welfare research, this translates to 'asking' animals about their motivation and emotional states—essentially what they want—through preference, learning, generalization and similar experiments [9,12,257]. A good example of such an approach is the analysis of positive and negative emotional states through optimistic or pessimistic cognitive biases when judging an ambiguous stimulus [89,103].

We think that computational models implementing functional mechanisms that account for internal subjective phenomena is a fruitful pathway. Unlike other fields such as behavioural ecology, we cannot rely on simple, elegant mathematical equations because the phenomenon of subjective cognition cannot be understood in isolation from the subject (the integrated phenotype [256]). It is complex and includes the interaction of many stochastic components, recursion, top-down causation and emergence. Agent-based simulation [267–269] with evolutionary adaptation [45,197,270] seems the most feasible option. The above discussion points to possible ways of developing a computational system that implements basic functional units of subjective phenomena: predictive, re-entrant and top-down processing, elementary self-awareness and global workspace in a simple mechanistic system.

The cognitive architecture implementing evolutionary adaptation [50,106] may then provide the basic component for larger and more complex digital twin models. Such models will help monitor and predict health, behaviour and subjective wellbeing of animals. This opens an exciting and challenging avenue for computational animal welfare science.

Ethics. This is a theoretical paper dealing with computational models aimed to develop new ways to improve animal welfare. We therefore expect no ethical issues.

Data accessibility. This is a theoretical paper that contains no data. Example computer code and documentation are available at https://ahamodel.uib.no.

Authors' contributions. J.G. and S.E. started the development of the theoretical ideas on the architectural models of decision-making more than 10 years ago. S.B. extended the ideas to psychology and cognition and has rewritten the computer codes. J.G. and T.S.K. then extended the framework to the animal wellbeing field. S.B. wrote the initial draft of the manuscript. J.G., S.E. and T.S.K. provided theoretical developments and contributed critically to the development of the manuscript.

Competing interests. We have no competing interests.

Funding. This research is supported by the University of Bergen, the Institute of Marine Research, the Research Council of Norway (grant no. FRIMEDBIO 239834) and contributes to the Centre for Digital Life Norway.

Acknowledgements. This paper is dedicated to the memory of Victoria Braithwaite. The authors have benefited from discussions with Bernard Baars, Victoria Braithwaite, Lars O.E. Ebbesson, Christian Jørgensen, Marc Mangel, Steve Railsback and Ivar Rønnestad. We thank three anonymous reviewers for valuable comments on an earlier draft.

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

# Brief glossary

**Agent**  an autonomous entity that is capable of adaptive, goal-directed behaviour.

**Awareness**  a cognitive state that results in the representation of an environmental or an internal object as a whole so as to create an isomorphism between this object and its subjective representation.

**Consciousness**  awareness of the agent's own existence and relationships with its environment.

**Experience**  a subjective process by which an agent perceives its external and internal environment through awareness. Note that we consider experience a broader concept than feeling that also includes intellectual experience, belief, etc.

**Feeling**  specific state of awareness closely linked with subjective experience, this can include sensory feelings and emotional feelings.

**Mind**  an intelligent computing system that implements learning, cognition and behavioural control. Mind can include a capacity for awareness and is supplemented by numerous automatic processing modules.

**Sentience**  the capacity to experience subjective feelings.

**Subjective** (processes/states)  internal processes and states of the organism that exist from the first-person point of view; their existence is inseparable from and cannot be defined independently of the experiencing organism.