## [Reviewer comments · Royal Society Open Science]

Review History

RSOS-200687.R0 (Original submission)

Review form: Reviewer 1

Is the manuscript scientifically sound in its present form?

Yes

Are the interpretations and conclusions justified by the results?

Yes

Is the language acceptable?

Yes

Do you have any ethical concerns with this paper?

No

Have you any concerns about statistical analyses in this paper?

No

Recommendation?

Major revision is needed (please make suggestions in comments)

Comments to the Author(s)

Very well written, interesting and ambitious paper, proposing what is essentially a new field of computational animal welfare science.

The front half of the paper is more conceptual and philosophical; I enjoyed this immensely and felt that it was extremely helpful and lucid, putting into words a variety of concepts which are familiar, but do not always appear together and are not always clearly linked.

The second half of their paper focuses more on the authors' own computational model. The features of this are interesting, and to me at least appear compelling, but this part of the paper is outside of my field of expertise so I am unable to comment in detail.

There is a potential issue of putting these elements together; as it is now, it is not clear if this is a conceptual review, or a description of a specific model and the resulting predictions. The authors will also need to check with the journal about the number of references used.

Specific comments:

Pg. 10, line 13: why model? Model for what?

Pg. 10, line 17: the self monitoring of well being sounds a bit esoteric and to my mind not completely necessary. I'd like to see a stronger argument for why this is essential. I agree that some type of self monitoring is important, but not self monitoring of overall well being – I'm not even sure if most human do a very good job with this.

Pg. 11, line 7: not clear what you mean by "avoiding dithering of GOS and behaviour" - rephrase.

Pg. 11, line 28: Yes, you can "start building computational models of wellbeing that integrate the organism's needs, motivation, emotion and subjective feeling", but you have not yet provided a frame work for how (and on what basis) these could be integrated?

Pg. 11, line 56: "Some of the empirical studies we will mention below were already known to [71] during the establishment of this mostly theoretically-inspired model. They cannot therefore be used as tests of the theory. Thus, we call them expectations rather than predictions." This is a start, but is it adequate. Sounds a bit circular. This will require some discussion.

Page 12 and 13: I found these 'expectations' really interesting, but the specific logic linking these to your model was not always clear. For example, the argument about displacement sounds reasonable, but I'm not sure what logic links this unambiguously to your model.

Pg. 13, line 29: "We argue that computer simulation of the animal's cognitive and behavioural function could provide a valuable tool to understand, monitor and improve the wellbeing status of animals" This sounds more like a hope, than a logical conclusion of what you have shown? I'd like to see more specific worked examples showing how the model get us here.

Pg. 14, line 14: "The previous excitement about neurobiological mechanisms of conscious experience seems reducing in the animal welfare field" - not sure what you mean by this - rephrase?

Review form: Reviewer 2

Is the manuscript scientifically sound in its present form?

Yes

Are the interpretations and conclusions justified by the results?

No

Is the language acceptable?

No

Do you have any ethical concerns with this paper?

No

Have you any concerns about statistical analyses in this paper?

No

Recommendation?

Major revision is needed (please make suggestions in comments)

Comments to the Author(s)

This is a well-written paper about a very interesting idea. I enjoyed reading it a lot. However, I think it falls short of what is required for a journal such as Open Science, in the following ways:

1. It presents an idea about how cognitive architectures might be used in a very important application. But it gives no evidence at all that they might indeed be useful or effective.
2. It is very short on detailed explanation of the idea itself. The curse of interdisciplinary writing is that one needs to explain oneself much more than in a specialist, single-discipline context. I can see the point of this paper because I work in computational cognitive architecture research, but I suspect most readers would simply not get the point. And even if they did get it, they would not be convinced because of the lack of evidence (see 1).
3. It contains no deeper insight than "here is a problem that one could solve with our cognitive architecture". For example, what would be the consequences for understanding of animal wellbeing? Would it be necessary to think about the problem in a particular way? Would new hypotheses be generated merely by the application itself? How would the modelling be done, and what data would it require? Does the architecture entail new hypotheses about the observed behaviour itself?
4. A specific thing that appears confused in the manuscript - and I'm not sure if it's the drafting or something deeper - is the reference to modularity. This is a very specific concept in cognitive science, as defined by Fodor, [2], included in the bibliography, but not cited at the point where modularity is introduced. Modularity is far from an agreed concept in cog sci, so it might be better to lay out the debate, and certainly to cite the relevant authors.

Minor issues:

Page 4 line 32: repeated "the"

Page 4 line 57: the metaphor "swarm of sensory input" is beautiful, but potentially misleading here, both to biologists and computer scientists, to whom the term is specific in different ways.

Various places: page citation format: e.g. [2, p. 51], not [2] p.51.

Page 11 line 57: "known to [71]" - I think the "to" is unintended here? (if not, the sentence is ill-formed.)

Page 12 line 32 "low-efficient" should be "low-efficiency"

Review form: Reviewer 3**Is the manuscript scientifically sound in its present form?**

Yes

Are the interpretations and conclusions justified by the results?

Yes

Is the language acceptable?

Yes

Do you have any ethical concerns with this paper?

No

Have you any concerns about statistical analyses in this paper?

No

Recommendation?

Accept with minor revision (please list in comments)

Comments to the Author(s)

This is an interesting theoretical paper that proposes the use of computational modelling in animal welfare assessment. It is very well written and argued and is significantly novel. I have a few minor comments/edits the authors should attend to before publication.

Abstract L36 generalisation AND decision- making

L38 I would argue we haven't made that much progress in human consciousness so how do we expect to understand this in beings we cannot communicate directly with. Marian Stamp

Dawkins argues that the issue of animal consciousness is actually hampering true progression in animal welfare - can you discuss (see her book *Why animals matter?*).

P4 L31 I find the whole idea of basing welfare on brain size a fallacy and I think "small-brained" should be removed. Animals lead very successful lives even with their smaller brains and are capable of complex tasks. Brain size is therefore not a valid measure.

P6 Although personality is discussed here I think the point that animals do exhibit personalities also termed boldness or proactive/reactive stress coping styles that may mean individuals of the same species behave differently. Thus, personality phenotype may constrain behavioural strategies which should be fed into your model.

P7 L13 environment AND satisfy.

P7 L14 you discuss biological functioning and feelings based welfare measures/definitions but not natural living. Can this be added?

P7 L32 add HPI (interrenal) for fish -of course HPA and HPI are vertebrates only. Should you mention other stress responses/hormones are activated in invertebrates?

P8 L4 We also need a means of knowing how much welfare is affected - the severity. Can your model do that?

P8 footnote - terms AS just what brains do.

General comments - throughout you use feel and feelings - I find this rather ambiguous so there is the sensory feel - to perceive something and then there is the emotional feel to subjectively experience. I think the term needs definition and I think experience is a better and more scientific word. You also use sentience but don't define it - I presume you are adopting Broom's definition? Perhaps it would be better to have a glossary or table of key terms so the non-specialist reader will be clear on your meaning?

P10 L24 So does an animal have to have some ability for sentience as outlined in Broom's definition or must it fulfill all of the criteria? Is there a line to be drawn? Or are you including all animals?

P13L3 onwards - you write as if animals can only do one thing at a time but they can do multiple things surely. For example a bird can fly and look for prey or predators so is using movement circuits and vision circuits? Does your model suggest only one circuit at a time so may be limited?

P14 L15 reducing in the animal welfare field - not sure what you mean here? Can you please rephrase?

As this is theoretical I take it you do not have an actual example of the model working. What would an empirical scientist need to know to use your model? Surely the internal state of an animal is impossible to know directly? Can you please discuss.

Decision letter (RSOS-200687.R0)

Dear Dr Budaev:

Manuscript ID RSOS-200687 entitled "Computational animal welfare: Towards cognitive architecture models of animal sentience, emotion and wellbeing" which you submitted to Royal Society Open Science, has been reviewed. The comments from reviewers are included at the bottom of this letter.

In view of the criticisms of the reviewers, the manuscript has been rejected in its current form. However, a new manuscript may be submitted which takes into consideration these comments.

Please note that resubmitting your manuscript does not guarantee eventual acceptance, and that your resubmission will be subject to peer review before a decision is made.

Your resubmitted manuscript should be submitted by 19-Nov-2020. If you are unable to submit by this date please contact the Editorial Office.

on behalf of Dr Joydeep Bhattacharya (Associate Editor) and Essi Viding (Subject Editor)
openscience@royalsociety.org

Reviewers' Comments to Author:

Reviewer: 1

Comments to the Author(s)

Very well written, interesting and ambitious paper, proposing what is essentially a new field of computational animal welfare science.

The front half of the paper is more conceptual and philosophical; I enjoyed this immensely and felt that it was extremely helpful and lucid, putting into words a variety of concepts which are familiar, but do not always appear together and are not always clearly linked.

The second half of their paper focuses more on the authors' own computational model. The features of this are interesting, and to me at least appear compelling, but this part of the paper is outside of my field of expertise so I am unable to comment in detail.

There is a potential issue of putting these elements together; as it is now, it is not clear if this is a conceptual review, or a description of a specific model and the resulting predictions. The authors will also need to check with the journal about the number of references used.

Specific comments:

Pg. 10, line 13: why model? Model for what?

Pg. 10, line 17: the self monitoring of well being sounds a bit esoteric and to my mind not completely necessary. I'd like to see a stronger argument for why this is essential. I agree that some type of self monitoring is important, but not self monitoring of overall well being — I'm not even sure if most human do a very good job with this.

Pg. 11, line 7: not clear what you mean by "avoiding dithering of GOS and behaviour" - rephrase.

Pg. 11, line 28: Yes, you can "start building computational models of wellbeing that integrate the organism's needs, motivation, emotion and subjective feeling", but you have not yet provided a frame work for how (and on what basis) these could be integrated?

Pg. 11, line 56: "Some of the empirical studies we will mention below were already known to [71] during the establishment of this mostly theoretically-inspired model. They cannot therefore be used as tests of the theory. Thus, we call them expectations rather than predictions." This is a start, but is it adequate. Sounds a bit circular. This will require some discussion.

Page 12 and 13: I found these 'expectations' really interesting, but the specific logic linking these to your model was not always clear. For example, the argument about displacement sounds reasonable, but I'm not sure what logic links this unambiguously to your model.

Pg. 13, line 29: "We argue that computer simulation of the animal's cognitive and behavioural function could provide a valuable tool to understand, monitor and improve the wellbeing status of animals" This sounds more like a hope, than a logical conclusion of what you have shown? I'd like to see more specific worked examples showing how the model get us here.

Pg. 14, line 14: "The previous excitement about neurobiological mechanisms of conscious experience seems reducing in the animal welfare field" - not sure what you mean by this - rephrase?

Reviewer: 2

Comments to the Author(s)

This is a well-written paper about a very interesting idea. I enjoyed reading it a lot. However, I think it falls short of what is required for a journal such as Open Science, in the following ways:

1. It presents an idea about how cognitive architectures might be used in a very important application. But it gives no evidence at all that they might indeed be useful or effective.
2. It is very short on detailed explanation of the idea itself. The curse of interdisciplinary writing is that one needs to explain oneself much more than in a specialist, single-discipline context. I can see the point of this paper because I work in computational cognitive architecture research, but I suspect most readers would simply not get the point. And even if they did get it, they would not be convinced because of the lack of evidence (see 1).
3. It contains no deeper insight than "here is a problem that one could solve with our cognitive architecture". For example, what would be the consequences for understanding of animal wellbeing? Would it be necessary to think about the problem in a particular way? Would new

hypotheses be generated merely by the application itself? How would the modelling be done, and what data would it require? Does the architecture entail new hypotheses about the observed behaviour itself?

4. A specific thing that appears confused in the manuscript - and I'm not sure if it's the drafting or something deeper - is the reference to modularity. This is a very specific concept in cognitive science, as defined by Fodor, [2], included in the bibliography, but not cited at the point where modularity is introduced. Modularity is far from an agreed concept in cog sci, so it might be better to lay out the debate, and certainly to cite the relevant authors.

Minor issues:

Page 4 line 32: repeated "the"

Page 4 line 57: the metaphor "swarm of sensory input" is beautiful, but potentially misleading here, both to biologists and computer scientists, to whom the term is specific in different ways.

Various places: page citation format: e.g. [2, p. 51], not [2] p.51.

Page 11 line 57: "known to [71]" - I think the "to" is unintended here? (if not, the sentence is ill-formed.)

Page 12 line 32 "low-efficient" should be "low-efficiency"

Reviewer: 3

Comments to the Author(s)

This is an interesting theoretical paper that proposes the use of computational modelling in animal welfare assessment. It is very well written and argued and is significantly novel. I have a few minor comments/edits the authors should attend to before publication.

Abstract L36 generalisation AND decision- making

L38 I would argue we haven't made that much progress in human consciousness so how do we expect to understand this in beings we cannot communicate directly with. Marian Stamp Dawkins argues that the issue of animal consciousness is actually hampering true progression in animal welfare - can you discuss (see her book Why animals matter?).

P4 L31 I find the whole idea of basing welfare on brain size a fallacy and I think "small-brained" should be removed. Animals lead very successful lives even with their smaller brains and are capable of complex tasks. Brain size is therefore not a valid measure.

P6 Although personality is discussed here I think the point that animals do exhibit personalities also termed boldness or proactive/reactive stress coping styles that may mean individuals of the same species behave differently. Thus, personality phenotype may constrain behavioural strategies which should be fed into your model.

P7 L13 environment AND satisfy.

P7 L14 you discuss biological functioning and feelings based welfare measures/definitions but not natural living. Can this be added?

P7 L32 add HPI (interrenal) for fish -of course HPA and HPI are vertebrates only. Should you mention other stress responses/hormones are activated in invertebrates?

P8 L4 We also need a means of knowing how much welfare is affected - the severity. Can your model do that?

P8 footnote - terms AS just what brains do.

General comments - throughout you use feel and feelings - I find this rather ambiguous so there is the sensory feel - to perceive something and then there is the emotional feel to subjectively experience. I think the term needs definition and I think experience is a better and more scientific word. You also use sentience but don't define it - I presume you are adopting Broom's definition? Perhaps it would be better to have a glossary or table of key terms so the non-specialist reader will be clear on your meaning?

P10 L24 So does an animal have to have some ability for sentience as outlined in Broom's definition or must it fulfill all of the criteria? Is there a line to be drawn? Or are you including all animals?

P13L3 onwards - you write as if animals can only do one thing at a time but they can do multiple things surely. For example a bird can fly and look for prey or predators so is using movement circuits and vision circuits? Does your model suggest only one circuit at a time so may be limited?

P14 L15 reducing in the animal welfare field - not sure what you mean here? Can you please rephrase?

As this is theoretical I take it you do not have an actual example of the model working. What would an empirical scientist need to know to use your model? Surely the internal state of an animal is impossible to know directly? Can you please discuss.

Author's Response to Decision Letter for (RSOS-200687.R0)

See Appendix A.

RSOS-201886.R0

Review form: Reviewer 1

Is the manuscript scientifically sound in its present form?

Yes

Are the interpretations and conclusions justified by the results?

Yes

Is the language acceptable?

Yes

Do you have any ethical concerns with this paper?

No

Have you any concerns about statistical analyses in this paper?

No

Recommendation?

Accept as is

Comments to the Author(s)

I am satisfied with these changes. No further changes are required.

Review form: Reviewer 2

Is the manuscript scientifically sound in its present form?

Yes

Are the interpretations and conclusions justified by the results?

Yes

Is the language acceptable?

Yes

Do you have any ethical concerns with this paper?

No

Have you any concerns about statistical analyses in this paper?

No

Recommendation?

Accept with minor revision (please list in comments)

Comments to the Author(s)

This revised paper is a substantial improvement on the previous version. The argument is much clearer, and the extra structure really helps us to see the point. I think there is a question that the editor must answer about whether the substantive novel contribution is adequate for OPEN SCIENCE. There is clearly a contribution here, but it seems to me not to be very large, and I am not sure that it will have great impact.

Minor issues

Throughout: "e.g." and "i.e." should be "e.g.," and "i.e.," respectively.

Throughout: OPEN SCIENCE uses numbering for second-level headings. Using this would make section references (particularly in the "structure of this paper" section) MUCH easier to follow.

Line 21: there seems to be a space in the middle of the word "Yet".

Line 53: Not clear what "stripping" means. I think you mean "stripping out" - that is, "removing".

L58: missing comma after [21]

L63: missing comma after [20,33]

L72: "a" is unnecessary here (technology is a mass noun in his usage).

L75-84: Please use subsection numbering (see above) and refer using it here. This section is really hard to read.

L96: missing comma after [51]

L130: "its"  "their" (wildebeest is plural here)

L166-7: (more serious point) The meaning of this sentence is unclear. As written it suggests that the animal will try to change its surroundings to match its internal predictions, but that is not usually the approach in predictive cognition. I can't suggest a wording because I don't know exactly what you mean.

L174-180: Modularity again. Now that you've tightened this section up, I'm not sure you really mean modularity in the Fodorian sense (which is very strict and absolute, but distinct from wetware), but more in the sense of small world networks (which you do seem to refer to explicitly). These two are really not the same thing. This paragraph is much clearer than before, but I still have a sense that it does not convey your intended meaning really clearly.

L381 "the" is unnecessary here.

L386: missing comma after [18,19,135]

L415: The first "The" (at the start of the sentence) is unnecessary.

L420: The first "The" (at the start of 4) is unnecessary.

L430: 2 missing commas (and they are really needed!): "in and the evolving gene pool adapts them to"  "in, and the evolving gene pool adapts them to,"

L451: Not clear what you mean here. If you mean your particular cognitive architecture, then omit "modelling". If you mean modelling using cognitive architectures in general, then omit "The".

L457: as above - if you mean your own CA, then replace "The" with "Our". If you mean in general, then replace "The" with "A".

L597: "simulations"  "simulation" or L598 "seems"  "seem" (syntactic number agreement).

L607: "Bernhard"  "Bernard".

Review form: Reviewer 3

Is the manuscript scientifically sound in its present form?

Yes

Are the interpretations and conclusions justified by the results?

Yes

Is the language acceptable?

Yes

Do you have any ethical concerns with this paper?

No

Have you any concerns about statistical analyses in this paper?

No

Recommendation?

Accept with minor revision (please list in comments)

Comments to the Author(s)

Reviewer 3

The authors have done an admirable job of revising their manuscript. I have a very minor suggestion before this is acceptable for publication:

If the authors do believe there is ample information on human consciousness they should provide citations so the reader can explore this.

Otherwise I am very happy to recommend publication.

Decision letter (RSOS-201886.R0)

Dear Dr Budaev

On behalf of the Editors, we are pleased to inform you that your Manuscript RSOS-201886 "Computational animal welfare: Towards cognitive architecture models of animal sentience, emotion and wellbeing" has been accepted for publication in Royal Society Open Science subject to minor revision in accordance with the referees' reports. Please find the referees' comments along with any feedback from the Editors below my signature.

Please submit your revised manuscript and required files (see below) no later than 7 days from today's (ie 12-Nov-2020) date. Note: the ScholarOne system will 'lock' if submission of the revision is attempted 7 or more days after the deadline. If you do not think you will be able to meet this deadline please contact the editorial office immediately.

on behalf of Dr Joydeep Bhattacharya (Associate Editor) and Essi Viding (Subject Editor)
openscience@royalsociety.org

Associate Editor Comments to Author (Dr Joydeep Bhattacharya):

Both reviewers and myself have found the revised version to be much improved from the previously submitted one. There is a remaining question on the lack of an adequate novelty; given the nature of the topic, I am sympathetic to the ideas as presented in your article and am pleased to recommend for publication. However, do note that this is subject to the final revision after incorporating the comments, all minor, left by the second reviewer. I am looking forward to seeing the final submission.

Reviewer comments to Author:

Reviewer: 1

Comments to the Author(s)

I am satisfied with these changes. No further changes are required.

Reviewer: 2

Comments to the Author(s)

This revised paper is a substantial improvement on the previous version. The argument is much clearer, and the extra structure really helps us to see the point. I think there is a question that the editor must answer about whether the substantive novel contribution is adequate for OPEN SCIENCE. There is clearly a contribution here, but it seems to me not to be very large, and I am not sure that it will have great impact.

Minor issues

Throughout: "e.g." and "i.e." should be "e.g.," and "i.e.," respectively.

Throughout: OPEN SCIENCE uses numbering for second-level headings. Using this would make section references (particularly in the "structure of this paper" section) MUCH easier to follow.

Line 21: there seems to be a space in the middle of the word "Yet".

Line 53: Not clear what "stripping" means. I think you mean "stripping out" - that is, "removing".

L58: missing comma after [21]

L63: missing comma after [20,33]

L72: "a" is unnecessary here (technology is a mass noun in his usage).

L75-84: Please use subsection numbering (see above) and refer using it here. This section is really hard to read.

L96: missing comma after [51]

L130: "its"  "their" (wildebeest is plural here)

L166-7: (more serious point) The meaning of this sentence is unclear. As written it suggests that the animal will try to change its surroundings to match its internal predictions, but that is not usually the approach in predictive cognition. I can't suggest a wording because I don't know exactly what you mean.

L174-180: Modularity again. Now that you've tightened this section up, I'm not sure you really mean modularity in the Fodorian sense (which is very strict and absolute, but distinct from wetware), but more in the sense of small world networks (which you do seem to refer to explicitly). These two are really not the same thing. This paragraph is much clearer than before, but I still have a sense that it does not convey your intended meaning really clearly.

L381 "the" is unnecessary here.

L386: missing comma after [18,19,135]

L415: The first "The" (at the start of the sentence) is unnecessary.

L420: The first "The" (at the start of 4) is unnecessary.

L430: 2 missing commas (and they are really needed!): "in and the evolving gene pool adapts them to"  "in, and the evolving gene pool adapts them to,"

L451: Not clear what you mean here. If you mean your particular cognitive architecture, then omit "modelling". If you mean modelling using cognitive architectures in general, then omit "The".

L457: as above - if you mean your own CA, then replace "The" with "Our". If you mean in general, then replace "The" with "A".

L597: "simulations"  "simulation" or L598 "seems"  "seem" (syntactic number agreement).

L607: "Bernhard"  "Bernard".

Reviewer: 3

Comments to the Author(s)

Reviewer 3

The authors have done an admirable job of revising their manuscript. I have a very minor suggestion before this is acceptable for publication:

If the authors do believe there is ample information on human consciousness they should provide citations so the reader can explore this.

Otherwise I am very happy to recommend publication.

===PREPARING YOUR MANUSCRIPT===

===PREPARING YOUR REVISION IN SCHOLARONE===

Author's Response to Decision Letter for (RSOS-201886.R0)

See Appendix B.

Decision letter (RSOS-201886.R1)

Dear Dr Budaev,

It is a pleasure to accept your manuscript entitled "Computational animal welfare: Towards cognitive architecture models of animal sentience, emotion and wellbeing" in its current form for publication in Royal Society Open Science. The comments of the reviewer(s) who reviewed your manuscript are included at the foot of this letter.

on behalf of Dr Joydeep Bhattacharya (Associate Editor) and Essi Viding (Subject Editor)

Appendix A

Dear Editor,

Thank you very much for allowing us to resubmit our manuscript. The reviewers' comments were indispensable for improving the paper. We have thoroughly revised it as suggested by the reviewers. Below is the description of the changes made.

There are so many changes that it is not easy to point to each specifically, but our responses to the reviewers follow their points consecutively. The main reviewers' points are marked by bold typeface and separated by double colon (we repeat the main reviewer's topic where appropriate). We refer to the line numbers in the manuscript revision that go throughout the whole text in the PDF draft generated by ScholarOne.

Sincerely,
Sergey Budaev

Reviewer 1

The paper combines a conceptual review and a description of a specific model framework. We revised the Introduction and especially its end to show that our model is a new proposal for the animal welfare area.

Specific comments: The text has been thoroughly rewritten and rephrased as suggested.

Reviewer 2

1. This paper describes a modelling framework that has been used in evolutionary ecology (e.g. Giske et al., 2013, 2014, Weidner et al., 2020, Jensen et al., 2020). There is also a similar applied model of appetite and feeding for use in aquaculture (unpublished, final stages of development). But its extension to the animal welfare area is at an early stage. This is now briefly explained at the end of Introduction with some references to previous work.

2. We have thoroughly rewritten the text, trying to make it easier to grasp. Also, the completely redrawn figure now abandons complex structure and provides a simple "outline." We also added a brief glossary (Revision lines 1286-1304).

3. This is a very valuable point that we previously missed. The manuscript now includes a new section "8. Consequences for behaviour and welfare" that addresses this (Revision line 546+).

4. We adapted the text on modularity along the lines suggested by the reviewer. Specifically, we note that modularity of the human cognitive architecture is debated and have added reference to Fodor (that was for some reasons missed), Revision line 180. The text now almost fully concerns general evolutionary/structural/ecological issues where there is more agreement than in cognitive science. We also decided not to extend the discussion about human cognitive modularity that will certainly distract most readers and unnecessarily inflate the text.

Minor issues:: We fixed all these and significantly rephrased the text.

Reviewer 3

We do not fully agree with the reviewer that there is little progress in the study of human consciousness. Its many neurobiological correlates are found, many theories are proposed, even computational models are developed. Some philosophers are, indeed, more sceptical about these

developments, but in our view this is one of the most exciting areas of research. We agree that there is a fundamental difficulty with non-verbal animals. But in animals, subjective processes can be inferred indirectly from the observed behaviour. Here computational models implementing alternative theories can be instrumental. One of the best ways to test a theory is to implement it into workable toy machine (even only a virtual simulation) that can run. Unfortunately, we could not allow the paper to grow further and therefore cannot extend the philosophical discussion. But we added a brief discussion on Marian Dawkins' book at the paragraph above (Revision lines 50-56). This will indeed provide a much more balanced introduction.

The brain size fallacy:: we agree and rephrased the text (Revision lines 104-106).

Personality:: we agree with the reviewer. One of the main inspirations for this modelling work was actually to account for personality (e.g. Giske et al., 2013, 2014) and individual differences in general (e.g. Jensen et al., 2020). But we unfortunately cannot add much more discussions here as it will cause a further growth of the paper. This is an interesting topic, so if the Editor decides it would be good to add an additional paragraph or so, we are happy to extend this point.

Natural living:: The text now refers to "Naturalness, i.e. ability to perform natural, ecologically adaptive behaviour ..." (Revision, line 233).

HPA axis:: We added reference to HPI in fish and noted specific stress hormones in invertebrates, e.g. hyperglycemic in crustacean (Revision lines 249-252). To avoid further growth of the text, we cannot provide more detailed discussion (especially on invertebrates, where recent studies are very interesting, e.g. hormonal mechanisms of stress and pain).

Severity of welfare effect:: a sentence on this was added (Revision line 332).

General comment on definitions:: we revised the text, which also caused refining some definitions. Our definitions are based on Broom, but we tried to make them as much as possible applicable to artificial agents (which is requirement of the modelling). To avoid confusion, we added a brief glossary of the main terms, see Revision lines 1286-1304 (incidentally, this is not an easy task, there is a diversity of views, so the terms/views could remain debatable).

So does an animal have to have some ability for sentience... or must it fulfil all of the criteria?::

The framework allows to add and combine different building blocks and therefore implement models of different species. This reflects the wide diversity and continuity of neurobehavioural architectures and capacities across taxa and species. It would be tempting to add more discussion on evolution (e.g. simplest stimulus-response organisms, organisms with competing neural modules adding capacity to choose and arbitrate, next top-down attention and so on up to a complex neurobiological architecture combining parallel competing modules, top-down control, subjective model, global broadcast etc.) but we have limited space.

Animals can only do one thing at a time:: Our writing was confusing, we now rephrased this. But for simplicity and tractability, the models assume that animal can do one type of activity at a time. We refer to this as "assumption that the animal performs one major activity at a time" (Revision lines 328-329). We feel that it is a less rigid and confusing wording. (Even though we have not implement parallelism because the main concern was so far on behavioural action selection, it is possible to run different kinds of circuits in parallel, e.g. vision and action selection).

Actual example of the model working:: This computational framework has been under development for several years and there were empirical modelling studies (e.g. Giske et al., 2013, 2014), as well as similar albeit simplified comparable architectural models for hormonal systems

(e.g. Weidner et al., 2020, Jensen et al., 2020). There is also an applied model of appetite and feeding for use in aquaculture (unpublished, final stages of development). But these were in different areas than animal welfare. We now clearly describe at the end of the Introduction that the extension of the framework to animal welfare is at an earlier stage (Revision lines 85-87).

What would an empirical scientist need to know to use your model? :: We have added a new section “8. Consequences for behaviour and welfare” to address these issues (Revision lines 546 and below).

Appendix B

Dear Editor,

Thank you very much for accepting our manuscript. We have revised it as suggested by the reviewers (most changes, rev. 2). We would like to take this opportunity to thank the reviewers for their valuable comments. This was a big help, not least, also correcting some glitches in English as none of us is native English speaker.

Reviewer 2:

We have now fixed all the issues noted by the second reviewer, as suggested. There are only two responses:

L166-7: (more serious point):: Yes, our wording is as intended, the animal tries to change its surroundings such as to match the environment to the internal prediction model. This indeed might seem counter-intuitive, like turning everything upside-down. We now added the name of this paradigm to the text: “active inference.”

L174-180: modularity:: We agree with the reviewer, we did not intend to use the modularity concept in the Fodorian sense (with (im)penetrability etc.), but mainly focused on the concept used in evo-devo/biology. We believe our generic sentence would be sufficient to direct the reader to specific literature and the debate without inflating the text.

Reviewer 3:

<<If the authors do believe there is ample information on human consciousness they should provide citations so the reader can explore this>>:: It is probably difficult to add more references to the text because it already is large and the bibliography is huge. We preferred references that could be reused in several places of the manuscript to avoid too much increase of the bibliography. In the revision, we changed “recent decade” to plural, which is more correct. We also added a reference to Banks’ “Encyclopedia of consciousness.” This is not the most recent source, but contains quite exhaustive overview of neuroscience and animal studies. We have already cited a few recent references (these left intact). If the Editor think desirable, we can add more references (Seth, Dehaene, Pockett, Irvine, Tononi, Noë, Friston etc.) at the cost of further bibliography increase.

We cannot disagree with the reviewer 3 here. It can be difficult to evaluate (and agree upon) how “big” progress is, the assessment is quite subjective and depends on the viewpoint. For some people, tackling into even basic neurobiological mechanisms of consciousness may be perceived as a huge advancement (compared to previously mainly speculative philosophical studies). But this progress, arguably, has not yet solved any major problem. So, it can be perceived by others as not really so strong, maybe initial steps.

We reformatted the text to the style used in the most recent issues of *Royal Society Open Science*, placing the figure caption to a separate file. We provided a file showing the differences between the initial submission and this revision (obtained by version comparison function of Microsoft Word).

Sincerely,

Sergey Budaev